# MDverse, shedding light on the dark matter of molecular dynamics simulations

Johanna KS Tiemann[1]*[†], Magdalena Szczuka[2], Lisa Bouarroudj[3], Mohamed Oussaren[3], Steven Garcia[4], Rebecca J Howard[5], Lucie Delemotte[6], Erik Lindahl[5,6], Marc Baaden[7], Kresten Lindorff-Larsen[1], Matthieu Chavent[2]*, Pierre Poulain[3]*

[1]Linderstrøm-Lang Centre for Protein Science, Department of Biology, University of Copenhagen, Copenhagen, Denmark; [2]Institut de Pharmacologie et Biologie Structurale, CNRS, Université de Toulouse, Toulouse, France; [3]Université Paris Cité, CNRS, Institut Jacques Monod, Paris, France; [4]Independent researcher, Amsterdam, Netherlands; [5]Department of Biochemistry and Biophysics, Science for Life Laboratory, Stockholm University, Stockholm, Sweden; [6]Department of applied physics, Science for Life Laboratory, KTH Royal Institute of Technology, Stockholm, Sweden; [7]Laboratoire de Biochimie Théorique, CNRS, Université Paris Cité, Paris, France

*For correspondence:
johanna.tiemann@gmail.com (JKST);
matthieu.chavent@ipbs.fr (MC);
pierre.poulain@u-paris.fr (PP)

Present address:
[†]NovozymesA/S, Lyngby, Denmark

**Abstract** The rise of open science and the absence of a global dedicated data repository for molecular dynamics (MD) simulations has led to the accumulation of MD files in generalist data repositories, constituting the *dark matter of MD* — data that is technically accessible, but neither indexed, curated, or easily searchable. Leveraging an original search strategy, we found and indexed about 250,000 files and 2000 datasets from Zenodo, Figshare and Open Science Framework. With a focus on files produced by the Gromacs MD software, we illustrate the potential offered by the mining of publicly available MD data. We identified systems with specific molecular composition and were able to characterize essential parameters of MD simulation such as temperature and simulation length, and could identify model resolution, such as all-atom and coarse-grain. Based on this analysis, we inferred metadata to propose a search engine prototype to explore the MD data. To continue in this direction, we call on the community to pursue the effort of sharing MD data, and to report and standardize metadata to reuse this valuable matter.

## eLife assessment

The study presents a **valuable** tool for searching molecular dynamics simulation data, making such datasets accessible for open science. The authors provide **convincing** evidence that it is possible to identify noteworthy molecular dynamics simulation datasets and that their analysis can produce information of value to the community.

## Introduction

The volume of data available in biology has increased tremendously (*Marx, 2013*; *Stephens et al., 2015*), through the emergence of high-throughput experimental technologies, often referred to as -omics, and the development of efficient computational techniques, associated with high-performance computing resources. The Open Access (OA) movement to make research results free and available to anyone (including e.g. the Budapest Open Access Initiative and the Berlin declaration on Open Access to Knowledge) has led to an explosive growth of research data made available by scientists (*Wilson*

*et al., 2021*). The FAIR (Findable, Accessible, Interoperable and Reusable) principles *Wilkinson et al., 2016* have emerged to structure the sharing of these data with the goals of reusing research data and to contribute to the scientific reproducibility. This leads to a world where research data has become widely available and exploitable, and consequently new applications based on artificial intelligence (AI) emerged. One example is AlphaFold (*Jumper et al., 2021*), which enables the construction of a structural model of any protein from its sequence. However, it is important to be aware that the development of AlphaFold was only possible because of the existence of extremely well annotated and cleaned open databases of protein structures (wwPDB *Berman et al., 2003*) and sequences (*UniProt Consortium, 2022*). Similarly, accurate predictions of NMR chemical shifts and chemical-shift-driven structure determination was only made possible via a community-driven collection of NMR data in the Biological Magnetic Resonance Data Bank (*Hoch et al., 2023*). One can easily imagine novel possibilities of AI and deep learning reusing previous research data in other fields, if that data is curated and made available at a large scale (*Fan and Shi, 2022*; *Mahmud et al., 2021*).

Molecular Dynamics (MD) is an example of a well-established research field where simulations give valuable insights into dynamic processes, ranging from biological phenomena to material science (*Perilla et al., 2015*; *Hollingsworth and Dror, 2018*; *Yoo et al., 2020*; *Alessandri et al., 2021*; *Krishna et al., 2021*). By unraveling motions at details and timescales invisible to the eye, this well-established technique complements numerous experimental approaches (*Bottaro and Lindorff-Larsen, 2018*; *Marklund and Benesch, 2019*; *Fawzi et al., 2021*). Nowadays, large amounts of MD data could be generated when modelling large molecular systems (*Gupta et al., 2022*) or when applying biased sampling methods (*Hénin et al., 2022*). Most of these simulations are performed to decipher specific molecular phenomena, but typically they are only used for a single publication. We have to confess that many of us used to believe that it was not worth the storage to collect all simulations (in particular since all might not have the same quality), but in hindsight this was wrong. Storage is exceptionally cheap compared to the resources used to generate simulations data, and they represent a potential goldmine of information for researchers wanting to reanalyze them (*Antila et al., 2021*), in particular when modern machine-learning methods are typically limited by the amount of training data. In the era of open and data-driven science, it is critical to render the data generated by MD simulations not only technically available but also practically usable by the scientific community. In this endeavor, discussions started a few years ago (*Abraham et al., 2019*; *Abriata et al., 2020*; *Merz et al., 2020*) and the MD data sharing trend has been accelerated with the effort of the MD community to release simulation results related to the COVID-19 pandemic (*Amaro and Mulholland, 2020*; *Mulholland and Amaro, 2020*) in a centralized database (https://covid.bioexcel.eu). Specific databases have also been developed to store sets of simulations related to protein structures (MoDEL: *Meyer et al., 2010*), membrane proteins in general (MemProtMD: *Stansfeld et al., 2015*; *Newport et al., 2019*), G-protein-coupled receptors in particular (GPCRmd: *Rodríguez-Espigares et al., 2020*), or lipids (Lipidbook: *Domański et al., 2010*, NMRLipids Databank: *Kiirikki et al., 2023*).

Albeit previous attempts in the past (*Tai et al., 2004*; *Meyer et al., 2010*), there is, as of now, no central data repository that could host all kinds of MD simulation files. This is not only due to the huge volume of data and its heterogeneity, but also because interoperability of the many file formats used adds to the complexity. Thus, faced with the deluge of biosimulation data (*Hospital et al., 2020*), researchers often share their simulation files in multiple generalist data repositories. This makes it difficult to search and find available data on, for example, a specific protein or a given set of parameters. We are qualifying this amount of scattered data as *the dark matter of MD*, and we believe it is essential to shed light onto this overlooked but high-potential volume of data. When unlocked, publicly available MD files will gain more visibility. This will help people to access and reuse these data more easily and overall, by making MD simulation data more FAIR (*Wilkinson et al., 2016*), it will also improve the reproducibility of MD simulations (*Elofsson et al., 2019*; *Porubsky et al., 2020*; *Bonomi et al., 2019*).

In this work, we have employed a search strategy to index scattered MD simulation files deposited in generalist data repositories. With a focus on the files generated by the Gromacs MD software, we performed a proof-of-concept large-scale analysis of publicly available MD data. We revealed the high value of these data and highlighted the different categories of the simulated molecules, as well as the biophysical conditions applied to these systems. Based on these results and our annotations, we proposed a search engine prototype to easily explore this *dark matter of MD*. Finally, building on

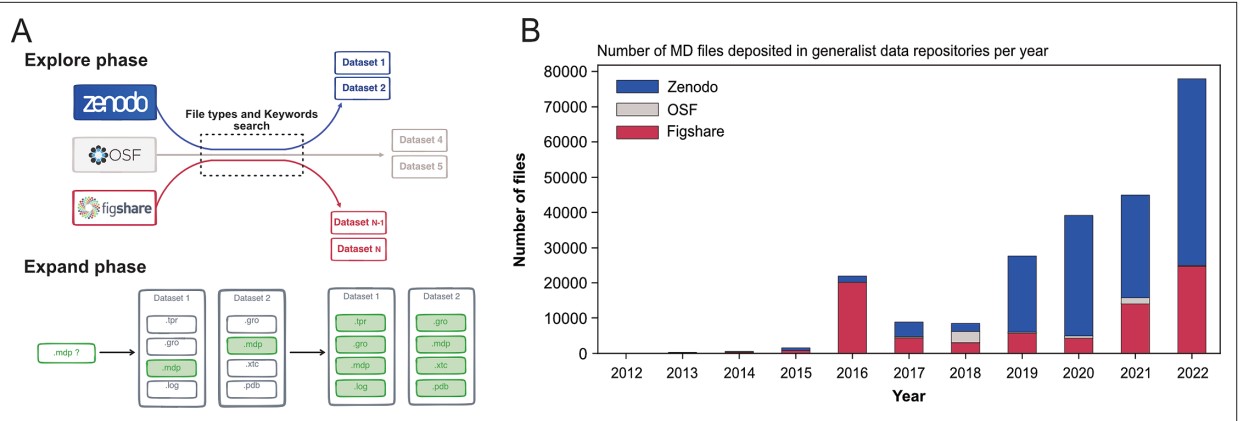

**Figure 1.** Explore and Expand ($Ex^2$) strategy used to index MD-related files and number of deposited files in generalist data repositories, identified by this strategy. (**A**) Explore and Expand ($Ex^2$) strategy used to index and collect MD-related files. Within the explore phase, we search in the respective data repositories for datasets that contain specific keywords (e.g. 'molecular dynamics', 'md simulation', 'namd', 'martini'...) in conjunction with specific file extensions (e.g. 'mdp', 'psf', 'parm7'...), depending on their uniqueness and level of trust to not report false-positives (i.e. not MD related). In the expand phase, the content of the identified datasets is fully cataloged, including files that individually could result in false positives (such as e.g. '.log' files). (**B**) Number of deposited files in generalist data repositories, identified by our $Ex^2$ strategy.

this experience, we provide simple guidelines for data sharing to gradually improve the FAIRness of MD data.

## Results

With the rise of open science, researchers increasingly share their data and deposit them into generalist data repositories, such as Zenodo (https://zenodo.org), Figshare (https://figshare.com), Open Science Framework (OSF, https://osf.io), and Dryad (https://datadryad.org/). In this first attempt to find out how many files related to MD are deposited in data repositories, we focused our exploration on three major data repositories: Figshare (~3.3 million files, ~112 TB of data, as of January 2023), OSF (~2 million files, as of November 2022) [Figures provided by Figshare and OSF user support teams.], and Zenodo (~9.9 million files, ~1.3 PB of data, as of December 2022; *Panero and Benito, 2022*).

One immediate strategy to index MD simulation files available in data repositories is to perform a text-based Google-like search. For that, one queries these repositories with keywords such as 'molecular dynamics' or 'Gromacs'. Unfortunately, we experienced many false positives with this search strategy. This could be explained by the strong discrepancy we observed in the quantity and quality of metadata (title, description) accompanying datasets and queried in text-based search. For instance, a description text could be composed of a couple of words to more than 1200 words. Metadata is provided by the user depositing the data, with no incentive to issue relevant details to support the understanding of the simulation. For the three data repositories studied, no human curation other by that of the providers is performed when submitting data. It is also worth mentioning that title and description are provided as free-text and do not abide to any controlled vocabulary such as a specific MD ontology.

**Table 1.** Statistics of the MD-related datasets and files found in the data repositories Figshare, OSF, and Zenodo.

| Data repository | datasets | first dataset | latest dataset | files | total size (GB) | zip files | files within zip | total files |
|---|---|---|---|---|---|---|---|---|
| Zenodo | 1011 | 19/11/2014 | 05/03/2023 | 20,250 | 12,851 | 1780 | 141,304 | 161,554 |
| Figshare | 913 | 20/08/2012 | 03/03/2023 | 3336 | 736 | 590 | 74,720 | 78,056 |
| OSF | 55 | 24/05/2017 | 05/02/2023 | 6146 | 495 | 14 | 0 | 6146 |
| *Total* | *1979* | – | – | *29,732* | *14,082* | *2384* | *216,024* | *245,756* |

To circumvent this issue, we developed an original and specific search strategy that we called *Explore and Expand* ($Ex^2$) (see *Figure 1A* and Materials and methods section) and that relies on a combination of file types and keywords queries. In the *Explore* phase, we searched for files based on their file types (for instance: .xtc, .gro, etc) with MD-related keywords (for instance: 'molecular dynamics', 'Gromacs', 'Martini', etc). Each of these hit files belonged to a dataset, which we further screened in the *Expand* phase. There, we indexed all files found in a dataset identified in the previous *Explore* phase with, this time, no restriction to the collected file types (see *Figure 1A* and details on the data scraping procedure in the Materials and methods section).

Globally, we indexed about 250,000 files and 2000 datasets that represented 14 TB of data deposited between August 2012 and March 2023 (see *Table 1*). One major difficulty were the numerous files stored in zipped archives, about seven times more than files steadily available in datasets (see *Table 1*). While this choice is very convenient for depositing the files (as one just needs to provide one big zip file to upload to the data repository server), it hinders the analysis of MD files as data repositories only provide a limited preview of the content of the zip archives and completely inhibits, for example, data streaming for remote analysis and visualization. Files within zip files are not indexed and cannot be searched individually. The use of zip archives also hampers the reusability of MD data, since a specific file cannot be downloaded individually. One has to download the entire zip archive (sometimes with a size up to several gigabytes) to extract the one file of interest.

The first dataset we found related to MD data that has been deposited in August 2012 in Figshare and corresponds to the work of *Fuller et al., 2012* (see *Table 1*) but we may consider the start of more substantial deposition of the MD data to be 2016 with more than 20,000 files deposited, mainly in Figshare (see *Figure 1B*). While the number of files deposited in Zenodo was first relatively limited, the last few years (2020–2022) saw a steep increase, passing from a few thousands files in 2018 to almost 50,000 files in 2022 (see *Figure 1B*). In 2018, the number of MD files deposited in OSF was similar to those in the two other data repositories, but did not take off as much as the other data repositories. Zenodo seems to be favored by the MD community since 2019, even though Figshare in 2022 also saw a sharp increase in deposited MD files. The preference for Zenodo could also be explained by the fact that it is a publicly funded repository developed under the European OpenAIRE program and operated by CERN (*European Organization For Nuclear Research, 2013*). Overall, the trend showed a rise of deposited data with a steep increase in 2022 (*Figure 1B*). We believe that this trend will continue in future years, which will lead to a greater amount of MD data available. It is thus urgent to deploy a strategy to index this vast amount of data, and to allow the MD community to easily explore and reuse such gigantic resource. The following describes what is already feasible in terms of meta analysis, in particular what types of data are deposited in data repositories and the simulation setup parameters used by MD experts that have deposited their data.

With our $Ex^2$ strategy (see *Figure 1A*), we assigned the deposited files to the MD packages: AMBER (*Salomon-Ferrer et al., 2013*), DESMOND (*Bowers et al., 2006*), Gromacs (*Berendsen et al., 1995*; *Abraham et al., 2015*), and NAMD/CHARMM (*Phillips et al., 2020*; *Brooks et al., 2009*), based on

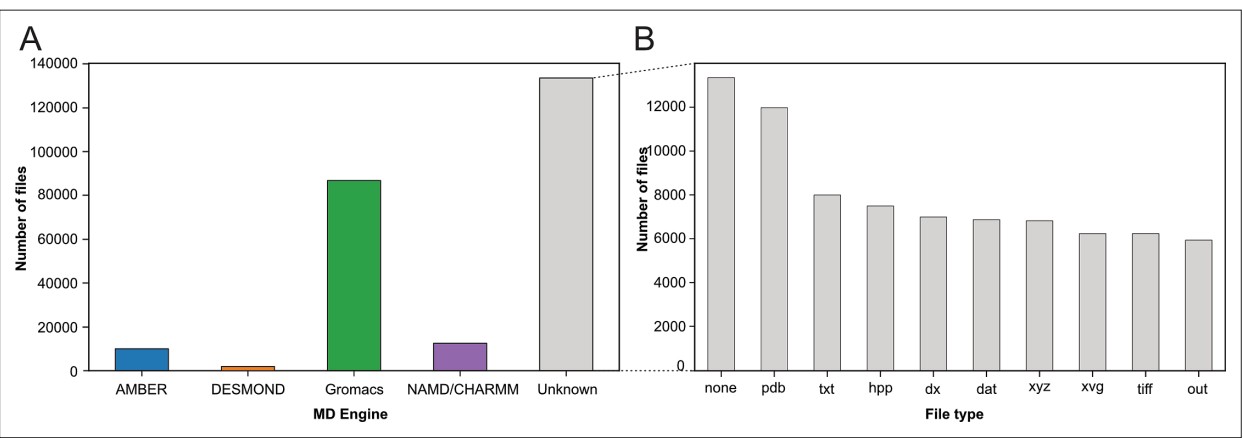

**Figure 2.** Categorization of index files based on their file types and assigned MD engine. (**A**) Distribution of files among MD simulation engines (**B**) Expansion of (**A**) MD Engine category 'Unknown' into the 10 most observed file types.

their corresponding file types (see Materials and methods section). In the case of NAMD/CHARMM, file extensions were mostly identical, which prevented us from distinguishing the respective files from these two MD programs. With 87,204 files deposited, the Gromacs program was most represented (see *Figure 2A*), followed by NAMD/CHARMM, AMBER, and DESMOND. This statistic is limited as it does not consider more specific databases related to a particular MD program. For example, the DE Shaw Research website contains a large amount of simulation data related to SARS-CoV-2 that has been generated using the ANTON supercomputer (https://www.deshawresearch.com/downloads/download_trajectory_sarscov2.cgi/) or other extensively simulated systems of interest to the community. However, this in itself might also serve as a good example, since few automated search strategies will be able to find custom stand-alone web servers as valuable repositories. Here, our goal was not to compare the availability of all data related to each MD program but to give a snapshot of the type of data available at a given time (i.e. March 2023) in generalist data repositories. Interestingly, many files (>133,000) were not directly associated to any MD program (see *Figure 2A* label 'Unknown'). We categorized these files based on their extensions (see *Figure 2B*). While 10% of these files were without file extension (*Figure 2B*, column *none*), we found numerous files corresponding to structure coordinates such as .pdb (~12,000) and .xyz (~6800) files. We also got images (.tiff files) and graphics (.xvg files). Finally, we found many text files such as .txt, .dat, and .out which can potentially hold details about how simulations were performed. Focusing further on files related to the Gromacs program, being currently most represented in the studied data repositories, we demonstrated in the following present possibilities to retrieve numerous information related to deposited MD simulations.

First, we were interested in what file types researchers deposited and thereby find potentially of great value to share. We therefore quantified the types of files generated by Gromacs (*Figure 3A*). The most represented file type is the.xtc file (28,559 files, representing 8.6 TB). This compressed (binary) file is used to store the trajectory of an MD simulation and is an important source of information to characterize the evolution of the simulated molecular system as a function of time. It is thus logical to mainly find this type of file shared in data repositories, as it is of great value for reusage and new analyses. Nevertheless, it is not directly readable but needs to be read by a third-party program, such as Gromacs itself, a molecular viewer like VMD (*Humphrey et al., 1996*) or an analysis library such as MDAnalysis (*Gowers et al., 2016*; *Michaud-Agrawal et al., 2011*). In addition, this trajectory file can only be of use in combination with a matching coordinates file, in order to correctly access the dynamics information stored in this file. Thus, as it is, this file is not easily mineable to extract useful information, especially if multiple .xtc and coordinate files are available in one dataset. Interestingly, we found 1406 .trr files, which contain trajectory but also additional information such as velocities, energy of the system, etc. While this file is especially useful in terms of reusability, the large size (can go up to several 100 GB) limits its deposition in most data repositories. For instance, a file cannot usually exceed 50 GB in Zenodo, 20 GB in Figshare (for free accounts) and 5 GB in OSF. Altogether, Gromacs trajectory files represented about 30,000 files in the three explored generalist repositories (34% of Gromacs files). This is a large number in comparison to existing trajectories stored in known databases dedicated to MD with 1700 MD trajectories available in MoDEL, 1737 trajectories (as of November 2022) available in GPCRmd, 5971 (as of January 2022) trajectories available in MemProtMD and 726 trajectories (as of March 2023) available in the NMRLipids Databank. Although fewer in count, these numbers correspond to manually or semi-automatically curated trajectories of specific systems, mostly proteins and lipids. Thus, ~30,000 MD trajectories available in generalist data repositories may represent a wider spectrum of simulated systems but need to be further analyzed and filtered to separate usable data from less interesting trajectories such as minimization or equilibration runs.

Given the large volume of data represented by .xtc files (see above), we could only scratch the surface of the information stored in these trajectory files by analyzing a subset of 779 .xtc files - one per dataset in which this type of file was found. We were able to get the size of the molecular systems and the number of frames available in these files (*Figure 3B*). The system size was up to more than one million atoms for a simulation of the TonB protein (*Virtanen et al., 2020*). The cumulative distribution of the number of frames showed that half of the files contain more than 10,000 frames. This conformational sampling can be very useful for other research fields besides the MD community that study, for instance, protein flexibility or protein engineering where diverse backbones can be of value. We found an .xtc file containing more than 5 million frames, where the authors probe the picosecond–nanosecond dynamics of T4 lysozyme and guide the MD simulation with NMR relaxation data (*Kümmerer*

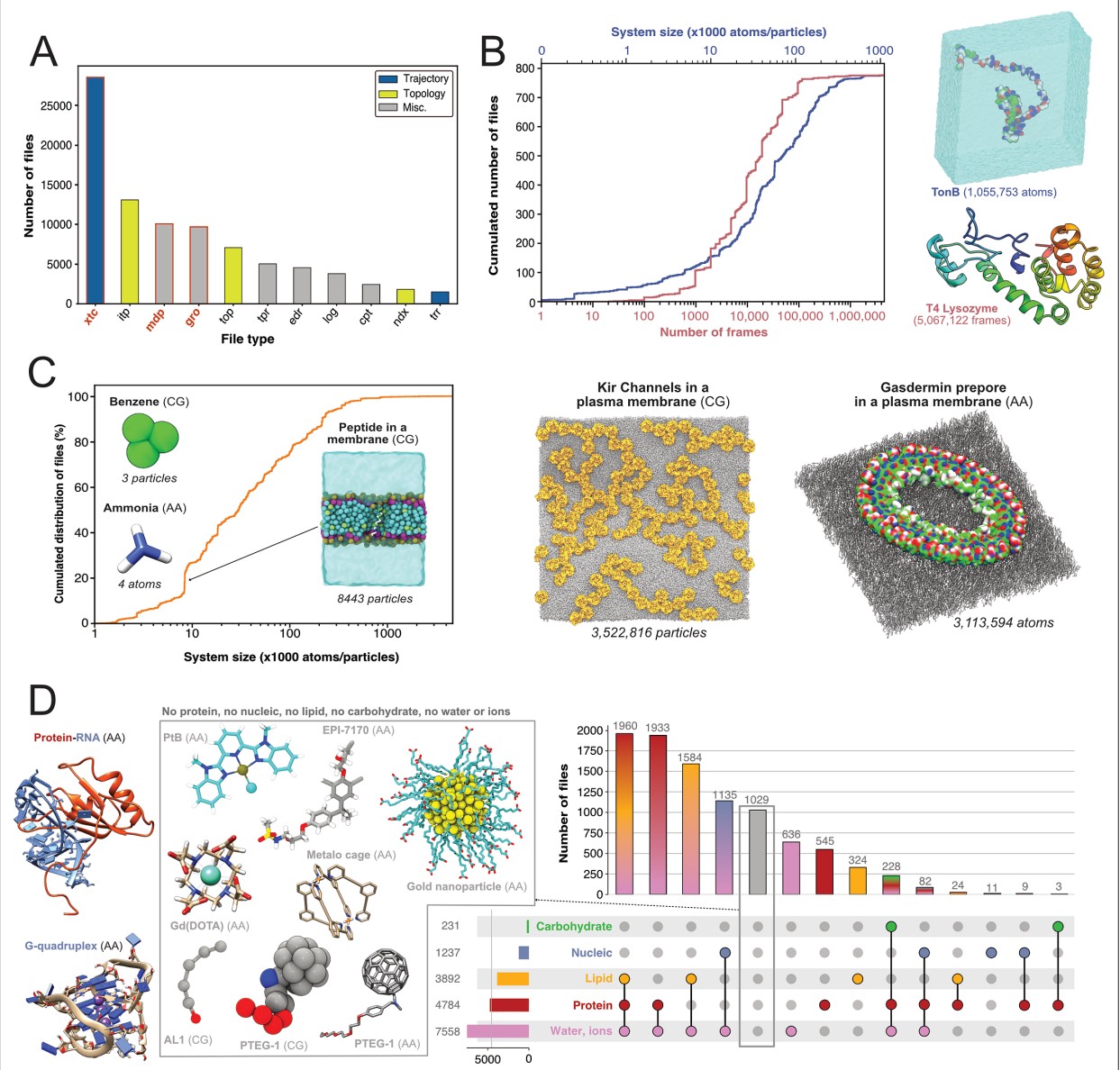

**Figure 3.** Content analysis of .xtc and .gro files. (**A**) Number of Gromacs-related files available in searched data repositories. In red, files used for further analyses. (**B**) Simple analyze of a subset of .xtc files with the cumulative distribution of the number of frames (in green) and the system size (in orange). (**C**) Cumulative distribution of the system sizes extracted from .gro files. (**D**) Upset plot of systems grouped by molecular composition, inferred from the analysis of .gro files. For this figure, 3D structures of representative systems were displayed, including soluble proteins such as TonB and T4 Lysozyme, membrane proteins such as Kir Channels and the Gasdermin prepore, Protein-/RNA and G-quadruplex and other non-protein molecules.

*et al., 2021*). Extending this analysis to all 28,559 .xtc files detected would be of great interest for a more holistic view, but this would require an initial step of careful checking and cleaning to be sure that these files are analyzable. Of note, as .xtc files also contain time stamps, it would be interesting to study the relationship between the time and the number of frames to get useful information about the sampling. Nevertheless, this analysis would be possible only for unbiased MD simulations. So, we would need to decipher if the .xtc file is coming from biased or unbiased simulations, which may not be trivial.

These results bring a first explanation on why there is not a single special-purpose repository for MD trajectory files. Databases dedicated to molecular structures such as the Protein Databank (*Berman et al., 2000*; *Kinjo et al., 2017*; *Armstrong et al., 2020*), or even the recent PDB-dev (*Burley et al., 2017*), designed for integrative models, cannot accept such large-size files, even less

if complete trajectories without reducing the number of frames would be uploaded. This would also require implementing extra steps of data curation and quality control. In addition, the size of the IT infrastructure and the human skills required for data curation represents a significant cost that could probably not be supported by a single institution.

Subsequently, our interest shifted towards exploring which systems are being investigated by MD researchers who deposit their files. We found 9718 .gro files which are text files that contain the number of particles and the Cartesian coordinates of the system modelled. By parsing the number of particles and the type of residue, we were able to give an overview of all Gromacs systems deposited (*Figure 3C, D*). In terms of system size, they ranged from very small - starting with two coarse-grain (CG) particles of graphite (*Piskorz et al., 2019*), followed by coordinates of a water molecule (3 atoms) (*Ivanov et al., 2017*), CG model of benzene (3 particles) (*Dandekar and Mondal, 2020*) and atomistic model of ammonia (4 atoms) (*Kelly and Smith, 2020*) — to go up to atomistic and coarse-grain systems composed of more than 3 million particles (*Duncan et al., 2020*; *Schaefer and Hummer, 2022*; *Figure 3C*). Interestingly, the system sizes in .gro files exceeded those of the analyzed .xtc files (*Figure 3B*). Even if we cannot exclude that the limited number of .xtc files analyzed (779 .xtc files selected from 28,559 .xtc files indexed) could explain this discrepancy, an alternate hypothesis is that the size of an .xtc file also depends on the number of frames stored. To reduce the size of .xtc files deposited in data repositories, besides removing some frames, researchers might also remove parts of the system, such as water molecules. As a consequence for reusability, this solvent removal could limit the number of suitable datasets available for researchers interested in re-analysing the simulation with respect to, in this case, water diffusion. While the size of systems extracted from .gro files was homogeneously spread, we observed a clear bump around system sizes of circa 8500 atoms/particles. This enrichment of data could be explained by the deposition of ~340 .gro files related to the simulation of a peptide translocation through a membrane (*Figure 3C*; *Kabelka et al., 2021*). Beyond 1 million particles/atoms, the number of systems is, for the moment, very limited.

We then analyzed residues in .gro files and inferred different types of molecular systems (see *Figure 3D*). Two of the most represented systems contained lipid molecules. This may be related to NMRLipids initiative (http://nmrlipids.blogspot.com). For several years, this consortium has been actively working on lipid modelling with a strong policy of data sharing and has contributed to share numerous datasets of membrane systems. As illustrated in *Figure 3C*, a variety of membrane systems, especially membrane proteins, were deposited. This highlights the vitality of this research field, and the will of this community to share their data. We also found numerous systems containing solvated proteins. This type of data, combined with .xtc trajectory files (see above), could be invaluable to describe protein dynamics and potentially train new artificial intelligence models to go beyond the current representation of the static protein structure (*Lane, 2023*). There was also a good proportion of systems containing nucleic acids alone or in interaction with proteins (1237 systems). At this time, we found only few systems containing carbohydrates that also contained proteins and corresponded to one study to model hyaluronan—CD44 interactions (*Vuorio et al., 2017*). Maybe a reason for this limited number is that systems containing sugars are often modelled using AMBER force field (*Salomon-Ferrer et al., 2013*), in combination with GLYCAM (*Kirschner et al., 2008*). A future study on the ~10,200 AMBER files deposited could retrieve more data related to carbohydrate containing systems. Given the current developments to model glycans (*Fadda, 2022*), we expect to see more deposited systems with carbohydrates in the coming years.

Finally, we found 1029 gro files which did not belong to the categories previously described. These files were mostly related to models of small molecules, or molecules used in organic chemistry (*Young et al., 2020*) and material science (*Piskorz et al., 2019*; *Zheng et al., 2022*) (see central panel, *Figure 3D*). Several datasets contained lists of small molecules used for calculating free energy of binding (*Aldeghi et al., 2016*), solubility of molecules (*Liu et al., 2016*), or osmotic coefficient (*Zhu, 2019*). Then, we identified models of nanoparticles (*Kyrychenko et al., 2012*; *Pohjolainen et al., 2016*), polymers (*Sarkar et al., 2020*; *Karunasena et al., 2021*; *Gertsen et al., 2020*), and drug molecules like EPI-7170, which binds disordered regions of proteins (*Zhu et al., 2022*). Finally, an interesting case from material sciences was the modelling of the PTEG-1 molecule, an addition of polar triethylene glycol (TEG) onto a fulleropyrrolidine molecule (see central panel, *Figure 3D*). This molecule was synthesized to improve semiconductors (*Jahani et al., 2014*). We found several models related to this peculiar molecule and its derivatives, both atomistic (*Qiu et al., 2017*; *Sami et al., 2022*) and coarse

grained (*Alessandri et al., 2020*). With a good indexing of data and appropriate metadata to identify modelled molecules, a simple search, which was previously to this study missing, could easily retrieve different models of the same molecule to compare them or to run multi-scale dynamics simulations. Beyond .gro files, we would like to analyze the ensemble of the ~12,000 .pdb extracted in this study (see *Figure 2B*) to better characterize the types of molecular structures deposited.

Another important category of deposited files are those containing information about the topology of the simulated molecules, including file extensions such as .itp and .top. Further, they are often the results of long parametrization processes (*Vanommeslaeghe and MacKerell, 2012*; *Souza et al., 2021*; *Wang et al., 2004*) and therefore of significant value for reusability. Based on our analysis, we indexed almost 20,000 topology files which could spare countless efforts to the MD community if these files could be easily found, annotated and reused. Interestingly, the number of .itp files was elevated (13,058 files) with a total size of 2 GB, while there were less .top files (7009 files) with a total size of 17 GB. Thus, .itp files seemed to contain much less information than the .top files. Among the remaining file types, .tpr files contain all the information to potentially directly run a simulation. Here, we found 4987 .tpr files, meaning that it could virtually be possible to rerun almost 5000 simulations without the burden of setting up the system to simulate. Finally, the 3730 .log files are also a source of useful information as it is relatively easy to parse this text file to extract details on how MD simulations were run, such as the version of Gromacs, which command line was used to run the simulation, etc.

Our next step was to gain insight into the parameter settings employed by the MD community, which may aid us in identifying preferences in MD setups and potential necessity for further education to avoid suboptimal or outdated configurations. We therefore analyzed 10,055 .mdp files stored in the different data repositories. These text files contain information regarding the input parameters to run the simulations such as the integrator, the number of steps, the different algorithms for barostat and thermostat, etc. (for more details see: https://manual.gromacs.org/documentation/current/user-guide/mdp-options.html).

We determined the expected simulation time corresponding to the product of two parameters found in .mdp files: the number of steps and the time step. Here, we acknowledge that one can set up a very long simulation time and stop the simulation before the end or, on contrary, use a limited time (especially when calculations are performed on HPC resources with wall-time) and then extend the simulation for a longer duration. Using only the .mdp file, we cannot know if the simulation reached its term. To do so, comparison with an .xtc file from the same dataset may help to answer this specific question. However, in this study, we were interested in MD setup practices, in particular what simulation time researchers would set up their system with - likely in the mindset to reach that ending time. We restricted this analysis to the 4623 .mdp files that used the *md* or *sd* integrator, and that have a simulation time above 1 ns. We found that the majority of the .mdp files were used for simulations of 50 ns or less (see *Figure 4A*). Further, 697 .mdp files with simulations times set-up between 50 ns and 1 µs and 585 .mdp files with simulation time above 1 µs were identified. As analyzing .gro files showed a good proportion of coarse-grained models (*Figure 3B, C*), we discriminated simulations setups for these two types of models using the time step as a simple cutoff. We considered that a time step greater than 10 fs (i.e. dt = 0.01) corresponded to MD setups for coarse grained models (*Ingólfsson et al., 2014*). Globally, we found that over all simulations, the setups for atomistic simulations were largely dominant. However, for simulations with a simulation time above 1 µs specifically, coarse-grain simulations represented 86% of all.

We then looked into the combinations of thermostat and barostat (see *Figure 4B*) from 9199 .mdp files. The main thermostat used is by far the V-rescale (*Bussi et al., 2007*) often associated with the Parrinello-Rahman barostat (*Parrinello and Rahman, 1981*). This thermostat was also used with the Berendsen barostat (*Berendsen et al., 1984*). In a few cases, we observed the use of the V-rescale thermostat with the very recently developed C-rescale barostat (*Bernetti and Bussi, 2020*). A total of 2021 .mdp files presented neither thermostat nor barostat, which means they would not be used in production runs. This could correspond to setups used for energy minimization, or to add ions to the system (with the genion command), or for molecular mechanics with Poisson–Boltzmann and surface area solvation (MM/PBSA) and molecular mechanics with generalised Born and surface area solvation (MM/GBSA) calculations (*Genheden and Ryde, 2015*).

Finally, we analyzed the range of starting temperatures used to perform simulations (see *Figure 4C*). We found a clear peak around the temperatures 298 K - 310 K which corresponds to the range

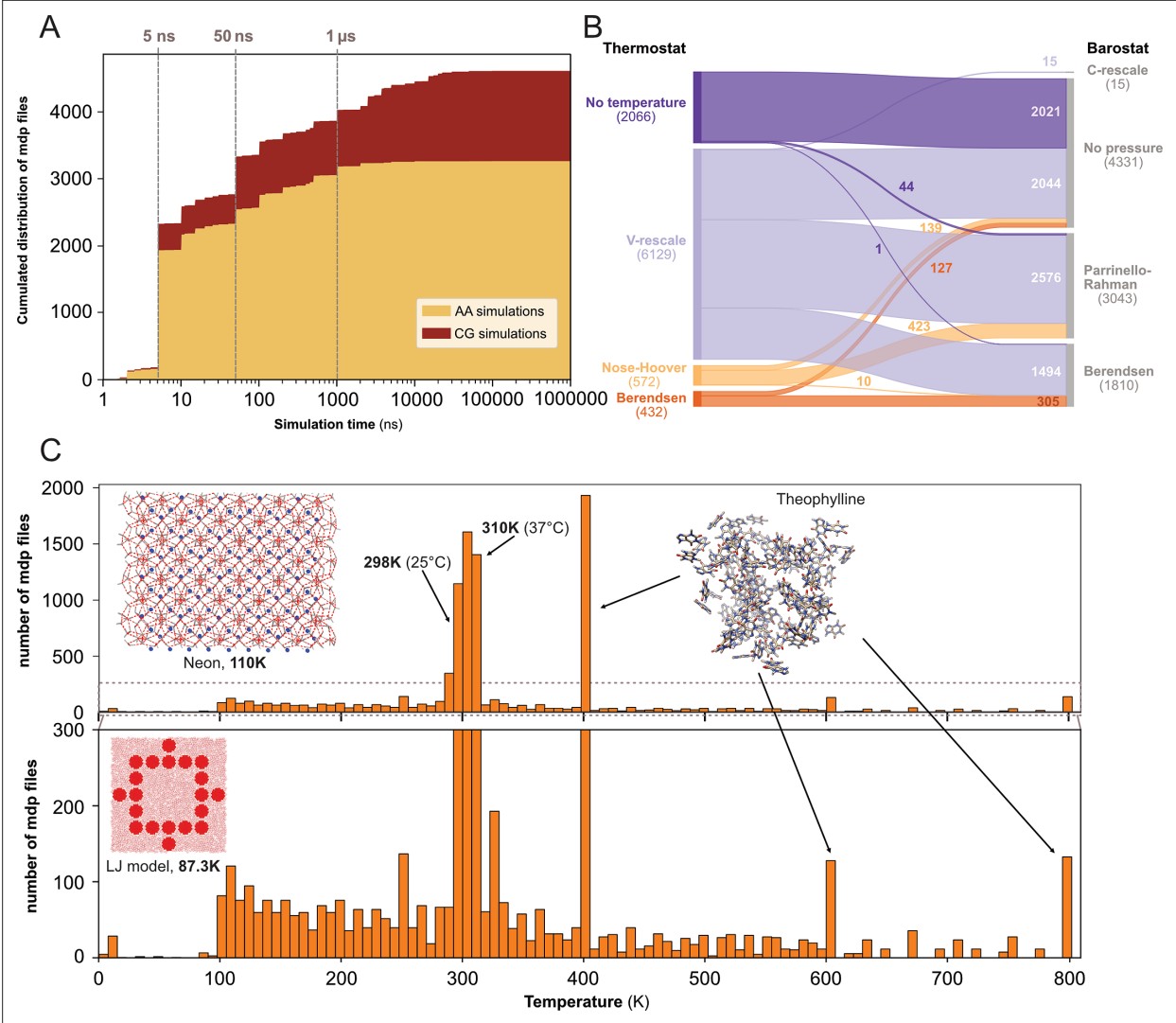

**Figure 4.** Content analysis of .mdp files. (**A**) Cumulative distribution of .mdp files versus the simulation time for all-atom and coarse-grain simulations. (**B**) Sankey graph of the repartition between different values for thermostat and barostat. (**C**) Temperature distribution, full scale in upper panel and zoom-in in lower panel.

between ambient room (298 K - 25 °C) and physiological (310 K - 37 °C) temperatures. Nevertheless, we also observed lower temperatures, which often relate to studies of specific organic systems or simulations of Lennard-Jones models (*Jeon et al., 2016*). Interestingly, we noticed the appearance of several pikes at 400 K, 600 K, and 800 K, which were not present before the end of the year 2022. These peaks corresponded to the same study related to the stability of hydrated crystals (*Dybeck et al., 2023*). Overall, this analysis revealed that a wide range of temperatures have been explored, starting mostly from 100 K and going up to 800 K.

To encourage further analysis of the collected files, we shared our data collection with the community in Zenodo (see Data availability statement). The data scrapping procedure and data analysis is available on GitHub with a detailed documentation. To let researchers having a quick glance and explore this data collection, we created a prototype web application called *MDverse data explorer* available at https://mdverse.streamlit.app/ and illustrated in *Figure 5A*. With this web application, it is easy to use keywords and filters to access interesting datasets for all MD engines, as well as .gro and .mdp files. Furthermore, when available, a description of the found data is provided and searchable for keywords (*Figure 5A*, on the left sidebar). The sets of data found can then be exported as a tab-separated values (.tsv) file for further analysis (*Figure 5B*).

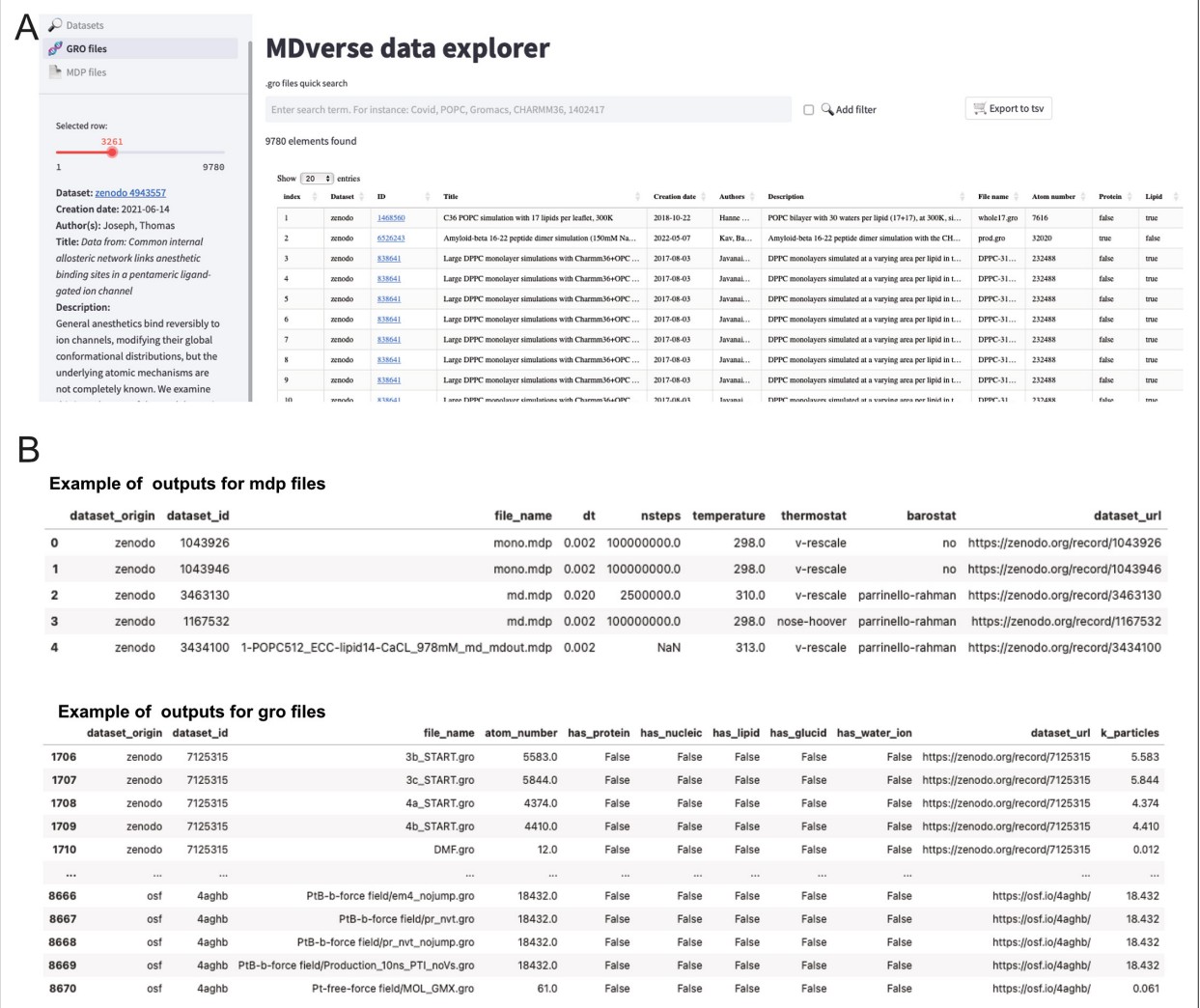

**Figure 5.** Snapshots of the MDverse data explorer, a prototype search engine to explore collected files and datasets. (**A**) General view of the web application. (**B**) Focus on the .mdp and .gro files sets of data exported as.tsv files. The web application also includes links to their original repository.

## Towards a better sharing of MD data

With this work, we have shown that it was possible to not only retrieve MD data from the generalist data repositories Zenodo, Figshare and OSF, but to shed light onto the *dark matter of MD* data in terms of learning current scientific practice, extracting valuable topology information, and analysing how the field is developing. Our objective was not to assess the quality of the data but only to show what kind of data was available. The $Ex^2$ strategy to find files related to MD simulations relied on the fact that many MD software output files with specific file extensions. This strategy could not be applied in research fields where data exhibits non-specific file types. We experienced this limitation while indexing zip archives related to MD simulations, where we were able to decide if a zip archive was pertinent for this work only by accessing the list of files contained in the archive. This valuable feature is provided by data repositories like Zenodo and Figshare, with some caveats, though.

As of March 2023, we managed to index 245,756 files from 1979 datasets, representing altogether 14 TB of data. This is a fraction of all files stored in data repositories. For instance, as of December 2022, Zenodo hosted about 9.9 million files for ~1.3 PB of data (*Panero and Benito, 2022*). All these files are stored on servers available 24/7. This high availability costs human resources, IT infrastructures and energy. Even if MD data represents only 1% of the total volume of data stored in Zenodo, we believe it is our responsibility, as a community, to develop a better sharing and reuse of MD simulation files - and it will neither have to be particularly cumbersome nor expensive. To this end,

we are proposing two solutions. First, improve practices for sharing and depositing MD data in data repositories. Second, improve the FAIRness of already available MD data notably by improving the quality of the current metadata.

## Guidelines for better sharing of MD simulation data

Without a community-approved methodology for depositing MD simulation files in data repositories, and based on the current experience we described here, we propose a few simple guidelines when sharing MD data to make them more FAIR (Findable, Accessible, Interoperable and Reusable):

- Avoid zip or tar archives whose content cannot be properly indexed by data repositories. As much as possible, deposit original data files directly.
- Describe the MD dataset with extensive metadata. Provide adequate information along your dataset, such as:

  The scope of the study, e.g. investigate conformation dynamics, benchmark force field,...

  The method on a basic (e.g. quantum mechanics, all-atom, coarse-grain) or advanced (accelerated, metadynamics, well-tempered) level.

  The MD software: name, version (tag) and whether modifications have been made.

  The simulation settings (for each of the steps, including minimization, equilibration and production): temperature(s), thermostat, barostat, time step, total runtime (simulation length), force field, additional force field parameters.

  The composition of the system, with the precise names of the molecules and their numbers, if possible also PDB, UniProt or Ensemble identifiers and whether the default structure has been modified.

  Give information about any post-processing of the uploaded files (e.g. truncation or stripping of the trajectory), including before and after values of what has been modified e.g. number of frames or number of atoms of uploaded files.

  Highlight especially valuable data, e.g. excessively QM-based parameterized molecules, and their parameter files.

Store this metadata in the description of the dataset. An adaptation of the Minimum Information About a Simulation Experiment (MIASE) guidelines *Waltemath et al., 2011* in the context of MD simulations would be useful to define required metadata.

- Link the MD dataset to other associated resources, such as:

  The research article (if any) for which these data have been produced. Datasets are usually mentioned in the research articles, but rarely the other way around, since the deposition has to be done prior to publication. However, it is eminently possible to submit a revised version, and providing a link to the related research paper in updated metadata of the MD dataset will ease the reference to the original publication upon data reuse.

  The code used to analyze the data, ideally deposited in the repository to guarantee availability, or in a GitHub or GitLab repository.

  Any other datasets that belong to the same study.

- Provide sufficient files to reproduce simulations and use a clear naming convention to make explicit links between related files. For instance, for the Gromacs MD engine, trajectory.xtc files could share the same names as structure.gro files (e.g. proteinA.gro and proteinA.xtc).
- Revisit your data deposition after paper acceptance and update information if necessary. Zenodo and Figshare provide a DOI for every new version of a dataset as well as a 'master' DOI that always refers to the latest version available.

These guidelines are complementary to the reliability and reproducibility checklist for molecular dynamics simulations (*Commun Biol, 2023*). Eventually, they could be implemented in machine actionable Data Management Plan (maDMP) (*Miksa et al., 2019*). So far, MD metadata is formalized as free text. We advocate for the creation of a standardized and controlled vocabulary to describe artifacts and properties of MD simulations. Normalized metadata will, in turn, enable scientific knowledge graphs (*Auer, 2018*; *Färber and Lamprecht, 2021*) that could link MD data, research articles and MD software in a rich network of research outputs.

Converging on a set of metadata and format requires a large consensus of different stakeholders, from users, to MD program developers, and journal editors. It would be especially useful to organize

specific workshops with representatives of all these communities to collectively tackle this specific issue.

## Improving metadata of current MD data

While indexing about 2000 MD datasets, we found that title and description accompanying these datasets were very heterogeneous in terms of quality and quantity and were difficult for machines to process automatically. It was sometimes impossible to find even basic information such as the identity of the molecular system simulated, the temperature or the length of the simulation. Without appropriate metadata, sharing data is pointless, and its reuse is doomed to fail (*Musen, 2022*). It is thus important to close the gap between the availability of MD data and its discoverability and description through appropriate metadata. We could gradually improve the metadata by following two strategies. First, since MD engines produce normalized and well-documented files, we could extract parameters of the simulation by parsing specific files. We already explored this path with Gromacs, by extracting the molecular size and composition from .gro files and the simulation time (with some limitations), thermostat and barostat from .mdp files. We could go even further, by extracting for instance Gromacs version from .log file (if provided) or by identifying the simulated system from its atomic topology stored in.gro files. This strategy can in principle be applied to files produced by other MD engines. A second approach that we are currently exploring uses data mining and named entity recognition (NER) methods (*Perera et al., 2020*) to automatically identify the molecular system, the temperature, and the simulation length from existing textual metadata (dataset title and description), providing they are of sufficient length. Finally, the possibilities afforded by large language models supplemented by domain-specific tools (*Bran et al., 2023*) might help interpret the heterogenous metadata that is often associated with the simulations.

## Future works

In the future, it is desirable to go further in terms of analysis and integrate other data repositories, such as Dryad and Dataverse instances (for example Recherche Data Gouv in France). The collaborative platform for source code GitHub could also be of interest. Albeit dedicated to source code and not designed to host large-size binary files, GitHub handles small to medium-size text files like tabular .csv and .tsv data files and has been extensively used to record cases of the Ebola epidemic in 2014 (*Perkel, 2016*) and the Covid-19 pandemic (*Johns Hopkins University, 2020*). Thus, GitHub could probably host small text-based MD simulation files. For Gromacs, we already found 70,000 parameter .mdp files and 55,000 structure .gro files. Scripts found along these files could also provide valuable insights to understand how a given MD analysis was performed. Finally, GitHub repositories might also be an entry point to find other datasets by linking to simulation data, such as institutional repositories (see for instance *Pesce and Lindorff-Larsen, 2023*). However, one potential point of concern is that repositories like GitHub or GitLab do not make any promises about long-term availability of repositories, in particular ones not under active development. Archiving of these repositories could be achieved in Zenodo (for data-centric repositories) or Software Heritage (*Di Cosmo and Zacchiroli, 2017*; for source-code-centric repositories).

An obvious next step is the enrichment of metadata with the hope to render open MD data more findable, accessible and ultimately reusable. Possible strategies have already been detailed previously in this paper. We could also go further by connecting MD data in the research ecosystem. For this, two apparent resources need to be linked to MD datasets: their associated research papers to mine more information and to establish a connection with the scientific context, and their simulated biomolecular systems, which ultimately could cross-reference MD datasets to reference databases such as *UniProt Consortium, 2022*, the PDB (*Berman et al., 2000*) or Lipid Maps (*Sud et al., 2007*). For already deposited datasets, the enrichment of metadata can only be achieved via systematic computational approaches, while for future depositions, a clear and uniformly used ontology and dedicated metadata reference file (as it is used by the PLUMED-NEST: *Bonomi et al., 2019*) would facilitate this task.

Eventually, front-end solutions such as the MDverse data explorer tool can evolve to being more user-friendly by interfacing the structures and dynamics with interactive 3D molecular viewers (*Tiemann et al., 2017*; *Kampfrath et al., 2022*; *Martinez and Baaden, 2021*).

## Conclusion

In this work, we showed that sharing data generated from MD simulations is now a common practice. From Zenodo, Figshare and OSF alone, we indexed about 250,000 files from 2000 datasets, and we showed that this trend is increasing. This data brings incentive and opportunities at different levels. First, for researchers who cannot access high-performance computing (HPC) facilities, or do not want to rerun a costly simulation to save time and energy, simulations of many systems are already available. These simulations could be useful to reanalyze existing trajectories, to extend simulations with already equilibrated systems or to compare simulations of a dedicated molecular system modelled with different settings. Second, building annotated and highly curated datasets for artificial intelligence will be invaluable to develop dynamic generative deep-learning models. Then, improving metadata along available data will foster their reuse and will mechanically increase the reproducibility of MD simulations. At last, we see here the occasion to push for good practices in the setup and production of MD simulations.

## Materials and methods

### Initial data collection

We searched for MD-related files in the data repositories Zenodo, Figshare and Open Science Framework (OSF). Queries were designed with a combination of file types and optionally keywords, depending on how a given file type was solely associated to MD simulations. We therefore built a list of manually curated and cross-checked file types and keywords (https://github.com/MDverse/mdws/blob/main/params/query.yml; *Poulain et al., 2023*). All queries were automated by Python scripts that utilized Application Programming Interfaces (APIs) provided by data repositories. Since APIs offered by data repositories were different, all implementations were performed in dedicated Python (*van Rossum, 1995*) (version 3.9.16) scripts with the NumPy (*Oliphant, 2007*) (version 1.24.2), Pandas (*McKinney, 2010*) (version 1.5.3) and Requests (version 2.28.2) libraries.

We made the assumption that files deposited by researchers in data repositories were coherent and all related to a same research project. Therefore, when an MD-related file was found in a dataset, all files belonging to this dataset were indexed, regardless of whether their file types were actually identified as MD simulation files. This is the core of the Explore and Expand strategy ($Ex^2$) we applied in this work and illustrated in *Figure 1*. By default, the last version of the datasets was collected.

When a zip file was found in a dataset, its content was extracted from a preview provided by Zenodo and Figshare. This preview was not provided through APIs, but as HTML code, which we parsed using the Beautiful Soup library (version 4.11.2). Note that the zip file preview for Zenodo was limited to the first 1000 files. To avoid false-positive files collected from zip archives, a final cleaning step was performed to remove all datasets that did not share at least one file type with the file type list mentioned above. In the case of OSF, there was no preview for zip files, so their content has not been retrieved.

### Gromacs files

After the initial data collection, Gromacs .mdp and .gro files were downloaded with the Pooch library (version 1.6.0). When a .mdp or .gro file was found to be in a zip archive, the latter was downloaded and the targeted .mdp or .gro file was selectively extracted from the archive. The same procedure was applied for a subset of .xtc files that consisted of about one .xtc file per Gromacs datasets.

Once downloaded, .mdp files were parsed to extract the following parameters: integrator, time step, number of steps, temperature, thermostat, and barostat. Values for thermostat and barostat were normalized according to values provided by the Gromacs documentation. For the simulation time analysis, we selected .mdp files with the *md* or *sd* integrator and with simulation time above 1 ns to exclude most minimization and equilibrating simulations. For the thermostat and barostat analysis, only files with non-missing values and with values listed in the Gromacs documentation were considered.

The .gro files were parsed with the MDAnalysis library (*Michaud-Agrawal et al., 2011*) to extract the number of particles of the system. Values found in the residue name column were also extracted and compared to a list of residues we manually associated to the following categories: protein,

lipid, nucleic acid, glucid and water or ions (https://github.com/MDverse/mdws/blob/main/params/residue_names.yml; *Poulain et al., 2023*).

The .xtc files were analyzed using the gmxcheck command (https://manual.gromacs.org/current/onlinehelp/gmx-check.html) to extract the number of particles and the number of frames.

## MDverse data explorer web app

The MDverse data explorer web application was built in Python with the Streamlit library. Data was downloaded from Zenodo (see the Data availability statement).

## System visualization and molecular graphics

Molecular graphics were performed with VMD (*Humphrey et al., 1996*) and Chimera (*Pettersen et al., 2004*). For all visualizations, .gro files containing molecular structure were used. In the case of the two structures in *Figure 3B*, .xtc files were manually assigned to their corresponding .gro (for the TonB protein) or .tpr (for the T4 Lysozyme) files based on their names in their datasets.

Origin of the structures displayed in this work:

### TonB

Dataset URL: https://zenodo.org/record/3756664
Publication (DOI): https://doi.org/10.1039/D0CP03473H

### T4 Lyzozyme

Dataset URL: https://zenodo.org/record/3989044
Publication (DOI): https://doi.org/10.1021/acs.jctc.0c01338

### Benzene

Dataset URL: https://figshare.com/articles/dataset/Capturing_Protein_Ligand_Recognition_Pathways_in_Coarse-Grained_Simulation/12517490/1
Publication (DOI): https://doi.org/10.1021/acs.jpclett.0c01683

### Ammonia

Dataset URL: https://figshare.com/articles/dataset/Alchemical_Hydration_Free-Energy_Calculations_Using_Molecular_Dynamics_with_Explicit_Polarization_and_Induced_Polarity_Decoupling_An_On_the_Fly_Polarization_Approach/11702442
Publication (DOI): https://doi.org/10.1021/acs.jctc.9b01139

### Peptide with membrane

Dataset URL: https://zenodo.org/record/4371296
Publication (DOI): https://doi.org/10.1021/acs.jcim.0c01312

### Kir channels

Dataset URL: https://zenodo.org/record/3634884
Publication (DOI): https://doi.org/10.1073/pnas.1918387117

### Gasdermin

Dataset URL: https://zenodo.org/record/6797842
Publication (DOI): https://doi.org/10.7554/eLife.81432

### Protein-RNA

Dataset URL: https://zenodo.org/record/1308045
Publication (DOI): https://doi.org/10.1371/journal.pcbi.1006642

### G-quadruplex

Dataset URL: https://zenodo.org/record/5594466
Publication (DOI): https://doi.org/10.1021/jacs.1c11248

### Ptb

Dataset URL: https://osf.io/4aghb/
Publication (DOI): https://doi.org/10.1073/pnas.2116543119

### EPI-7170

Dataset URL: https://zenodo.org/record/7120845
Publication (DOI): https://doi.org/10.1038/s41467-022-34077-z

### Gold nanoparticle

Dataset URL: https://acs.figshare.com/articles/dataset/Fluorescence_Probing_of_Thiol_Functionalized_Gold_Nanoparticles_Is_Alkylthiol_Coating_of_a_Nanoparticle_as_Hydrophobic_as_Expected_/2481241
Publication (DOI): https://doi.org/10.1021/jp3060813

### Gd(DOTA)

Dataset URL: https://acs.figshare.com/articles/dataset/Modeling_Gd_sup_3_sup_Complexes_for_Molecular_Dynamics_Simulations_Toward_a_Rational_Optimization_of_MRI_Contrast_Agents/20334621
Publication (DOI): https://doi.org/10.1021/acs.inorgchem.2c01597

### Metalo cage

Dataset URL: https://acs.figshare.com/articles/dataset/Rationalizing_the_Activity_of_an_Artificial_Diels-Alderase_Establishing_Efficient_and_Accurate_Protocols_for_Calculating_Supramolecular_Catalysis/11569452
Publication (DOI): https://doi.org/10.1021/jacs.9b10302

### AL1

Dataset URL: https://acs.figshare.com/articles/dataset/Nucleation_Mechanisms_of_Self-Assembled_Physisorbed_Monolayers_on_Graphite/8846045
Publication (DOI): https://doi.org/10.1021/acs.jpcc.9b01234

### PTEG-1 (all-atom)

Dataset URL: https://figshare.com/articles/dataset/PTEG-1_PP_and_N-DMBI_atomistic_force_fields/5458144
Publication (DOI): https://doi.org/10.1039/C7TA06609K

## PTEG-1 (coarse-grain)

Dataset URL: https://figshare.com/articles/dataset/Neat_and_P3HT-Based_Blend_Morphologies_for_PCBM_and_PTEG-1/12338633
Publication (DOI): https://doi.org/10.1002/adfm.202004799

## Theophylline

Dataset URL: https://figshare.com/articles/dataset/A_Comparison_of_Methods_for_Computing_Relative_Anhydrous_Hydrate_Stability_with_Molecular_Simulation/21644393
Publication (DOI): https://doi.org/10.1021/acs.cgd.2c00832

## Acknowledgements

We thank Lauri Mesilaakso, Bryan White, Jorge Hernansanz Biel, Zihwei Li and Kirill Baranov for their participation in the Copenhagen BioHackathon 2020, whose results showcased the need for a more advanced search strategy. We acknowledge Massimiliano Bonomi, Giovanni Bussi, Patrick Fuchs and Elise Lehoux for helpful discussions and suggestions. We also thank the Zenodo, Figshare and OSF support teams for providing figures on the content of their respective data repository and for their help in using APIs. This work was supported by Institut français du Danemark (Blåtand program, 2021), the Data Intelligence Institute of Paris (diiP, IdEx Université Paris Cité, ANR-18-IDEX-0001, 2023). JKST and KL-L acknowledge funding by the Novo Nordisk Foundation [NNF18OC0033950 to KL-L], and workshops funded by The BioExcel Center-of-Excellence (grant agreements 823830, 101093290).

## Additional information

### Competing interests

Lucie Delemotte: Reviewing editor, *eLife*. The other authors declare that no competing interests exist.

### Funding

| Funder | Grant reference number | Author |
| --- | --- | --- |
| Institut francais du Danemark | Blatand program 2021 | Matthieu Chavent<br>Pierre Poulain |
| Data Intelligence Institute of Paris | IdEx Université Paris Cité ANR-18-IDEX-0001 2023 | Mohamed Oussaren<br>Pierre Poulain |
| Novo Nordisk Foundation | NNF18OC0033950 | Johanna KS Tiemann<br>Kresten Lindorff-Larsen |
| BioExcel Center-of-Excellence | 823830 | Erik Lindahl |
| BioExcel Center-of-Excellence | 101093290 | Erik Lindahl |

The funders had no role in study design, data collection and interpretation, or the decision to submit the work for publication.

### Author contributions

Johanna KS Tiemann, Conceptualization, Data curation, Software, Formal analysis, Supervision, Validation, Investigation, Visualization, Methodology, Writing – original draft, Writing – review and editing; Magdalena Szczuka, Lisa Bouarroudj, Mohamed Oussaren, Data curation, Software, Formal analysis, Visualization; Steven Garcia, Data curation, Software, Formal analysis; Rebecca J Howard, Lucie Delemotte, Erik Lindahl, Marc Baaden, Kresten Lindorff-Larsen, Conceptualization, Writing – original draft, Writing – review and editing; Matthieu Chavent, Pierre Poulain, Conceptualization, Data curation, Software, Formal analysis, Supervision, Funding acquisition, Validation, Investigation, Visualization, Methodology, Writing – original draft, Writing – review and editing

Author ORCIDs

Rebecca J Howard (ORCID) https://orcid.org/0000-0003-2049-3378
Erik Lindahl (ORCID) https://orcid.org/0000-0002-2734-2794
Marc Baaden (ORCID) https://orcid.org/0000-0001-6472-0486
Kresten Lindorff-Larsen (ORCID) https://orcid.org/0000-0002-4750-6039
Matthieu Chavent (ORCID) https://orcid.org/0000-0003-4524-4773
Pierre Poulain (ORCID) https://orcid.org/0000-0003-4177-3619

Reviewer #1 (Public Review): https://doi.org/10.7554/eLife.90061.3.sa1
Reviewer #2 (Public Review): https://doi.org/10.7554/eLife.90061.3.sa2
Reviewer #3 (Public Review): https://doi.org/10.7554/eLife.90061.3.sa3
Author response https://doi.org/10.7554/eLife.90061.3.sa4

## Additional files

### Supplementary files

• MDAR checklist

### Data availability

Data files produced from the data collection and processing are shared in Parquet format in Zenodo. They are freely available under the Creative Commons Attribution 4.0 International license (CC-BY). Python scripts to search and index MD files, and to download and parse .mdp and .gro files are open-source (under the AGPL-3.0 license), freely available on GitHub and archived in Software Heritage (*Poulain et al., 2023*). A detailed documentation is provided along the scripts to easily reproduce the data collection and processing. Jupyter notebooks used to analyze results and create the figures of this paper are open-source (under the BSD 3-Clause license), freely available on GitHub and archived in Software Heritage (*Poulain, 2023*). The code of the MDverse data explorer web application is open-source (under the BSD 3-Clause license), freely available on GitHub and archived in Software Heritage (*Poulain and Oussaren, 2023*).

The following dataset was generated:

| Author(s) | Year | Dataset title | Dataset URL | Database and Identifier |
|---|---|---|---|---|
| Johanna KST, Mathieu C, Pierre P | 2023 | MDverse datasets | https://zenodo.org/records/7856806 | Zenodo, 10.5281/zenodo.7856806 |

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
