## [Editor Report · eLife assessment]

The study presents a **valuable** tool for searching molecular dynamics simulation data, making such datasets accessible for open science. The authors provide **convincing** evidence that it is possible to identify noteworthy molecular dynamics simulation datasets and that their analysis can produce information of value to the community.

---

## [Referee Report · Reviewer #1 (Public Review)]

Summary:

Tiemann et al. have undertaken an original study on the availability of molecular dynamics (MD) simulation datasets across the Internet. There is a widespread belief that extensive, well-curated MD datasets would enable the development of novel classes of AI models for structural biology. However, currently, there is no standard for sharing MD datasets. As generating MD datasets is energy-intensive, it is also important to facilitate the reuse of MD datasets to minimize energy consumption. Developing a universally accepted standard for depositing and curating MD datasets is a huge undertaking. The study by Tiemann et al. will be very valuable in informing policy developments toward this goal.

Strengths:

The study presents an original approach to addressing a growing concern in the field. It is clear that adopting a more collaborative approach could significantly enhance the impact of MD simulations in modern molecular sciences.

The timing of the work is appropriate, given the current interest in developing AI models for describing biomolecular dynamics.

Weaknesses:

The study primarily focuses on one major MD engine (GROMACS), although this limitation is not significant considering the proof-of-concept nature of the study.

---

## [Referee Report · Reviewer #2 (Public Review)]

Summary:

Molecular dynamics (MD) data is deposited in public, non-specialist repositories. This work starts from the premise that these data are a valuable resource as they could be used by other researchers to extract additional insights from these simulations; it could also potentially be used as training data for ML/AI approaches. The problem is that mining these data is difficult because they are not easy to find and work with. The primary goal of the authors was to discover and index these difficult-to-find MD datasets, which they call the "dark matter of the MD universe" (in contrast to data sets held in specialist databases).

The authors developed a search strategy that avoided the use of ill-defined metadata but instead relied on the knowledge of the restricted set of file formats used in MD simulations as a true marker for the data they were looking for. Detection of MD data marked a data set as relevant with a follow-up indexing strategy of all associated content. This "explore-and-expand" strategy allowed the authors for the first time to provide a realistic census of the MD data in non-specialist repositories.

As a proof of principle, they analyzed a subset of the data (primarily related to simulations with the popular Gromacs MD package) to summarize the types of simulated systems (primarily biomolecular systems) and commonly used simulation settings.

Based on their experience they propose best practices for metadata provision to make MD data FAIR (findable, accessible, interoperable, reusable).

A prototype search engine that works on the indexed datasets is made publicly available. All data and code are made freely available as open source/open data.

Strengths:

- The novel search strategy is based on relevant data to identify full datasets instead of relying on metadata and thus is likely to have many true positives and few false positives.

- The paper provides a first glimpse at the potential hidden treasures of MD simulations and force field parametrizations of molecules.

- Analysis of parameter settings of MD simulations from how researchers *actually* run simulations can provide valuable feedback to MD code developers for how to document/educate users. This approach is much better than analyzing what authors write in the Methods sections.

- The authors make a prototype search engine available.

- The guidelines for FAIR MD data are based on experience gained from trying to make sense of the data.

Weaknesses:

- So far the work is a proof-of-concept that focuses on MD data produced by Gromacs (which was prevalent under all indexed and identified packages).

As discussed in the manuscript, some types of biomolecules are likely underrepresented because different communities have different preferences for force fields/MD codes (for example: carbohydrates with AMBER/GLYCAM using AMBER MD instead of Gromacs).

- Materials sciences seem to be severely under-represented - commonly used codes in this area such as LAMMPS are not even detected, and only very few examples could be identified. As it is, the paper primarily provides an insight into the *biomolecular* MD simulation world.

The authors succeed in providing a first realistic view on what MD data is available in public repositories. In particular, their explore-expand approach has the potential to be customized for all kinds of specialist simulation data, whereby specific artifacts are

used as fiducial markers instead of metadata. The more detailed analysis is limited to Gromacs simulations and primarily biomolecular simulations (even though MD is also widely used in other fields such as the materials sciences). This restricted view may simply be correlated with the user community of Gromacs and hopefully, follow-up studies from this work will shed more light on this shortcoming.

The study quantified the number of trajectories currently held in structured databases as ~10k vs ~30k in generalist repositories. To go beyond the proof-of-principle analysis it would be interesting to analyze the data in specialist repositories in the same way as the one in the generalist ones, especially as there are now efforts underway to create a database for MD simulations (Grant 'Molecular dynamics simulation for biology and chemistry research' to establish MDDB' DOI 10.3030/101094651). One should note that structured databases do not invalidate the approach pioneered in this work; if anything they are orthogonal to each other and both will likely play an important role in growing the usefulness of MD simulations in the future.

---

## [Referee Report · Reviewer #3 (Public Review)]

Molecular dynamics (MD) simulations nowadays are an essential element of structural biology investigations, complementing experiments and aiding their interpretation by revealing transient processes or details (such as the effects of glycosylation on the SARS-CoV-2 spike protein, for example Casalino et al. ACS Cent. Sci. 2020; 6, 10, 1722-1734 https://doi.org/10.1021/acscentsci.0c01056) that cannot be observed directly. MD simulations can allow for the calculation of thermodynamic, kinetic, and other properties and the prediction of biological or chemical activity. MD simulations can now serve as "computational assays" (Huggins et al. WIREs Comput Mol Sci. 2019; 9:e1393. https://doi.org/10.1002/wcms.1393). Conceptually, MD simulations have played a crucial role in developing the understanding that the dynamics and conformational behaviour of biological macromolecules are essential to their function, and are shaped by evolution. Atomistic simulations range up to the billion atom scale with exascale resources (e.g. simulations of SARS-CoV-2 in a respiratory aerosol. Dommer et al. The International Journal of High Performance Computing Applications. 2023; 37:28-44. doi:10.1177/10943420221128233), while coarse-grained models allow simulations on even larger length- and timescales. Simulations with combined quantum mechanics/molecular mechanics (QM/MM) methods can investigate biochemical reactivity, and overcome limitations of empirical forcefields (Cui et al. J. Phys. Chem. B 2021; 125, 689 https://doi.org/10.1021/acs.jpcb.0c09898).

MD simulations generate large amounts of data (e.g. structures along the MD trajectory) and increasingly, e.g. because of funder mandates for open science, these data are deposited in publicly accessible repositories. There is real potential to learn from these data en masse, not only to understand biomolecular dynamics but also to explore methodological issues. Deposition of data is haphazard and lags far behind experimental structural biology, however, and it is also hard to answer the apparently simple question of "what is out there?". This is the question that Tiemann et al explore in this nice and important work, focusing on simulations run with the widely used GROMACS package. They develop a search strategy and identify almost 2,000 datasets from Zenodo, Figshare and Open Science Framework. This provides a very useful resource. For these datasets, they analyse features of the simulations (e.g. atomistic or coarse-grained), which provides a useful snapshot of current simulation approaches. The analysis is presented clearly and discussed insightfully. They also present a search engine to explore MD data, the MDverse data explorer, which promises to be a very useful tool.

As the authors state: "Eventually, front-end solutions such as the MDverse data explorer tool can evolve being more user-friendly by interfacing the structures and dynamics with interactive 3D molecular viewers". This will make MD simulations accessible to non-specialists and researchers in other areas. I would envisage that this will also include approaches using interactive virtual reality for an immersive exploration of structure and dynamics, and virtual collaboration (e.g. O'Connor et al., Sci. Adv.4, eaat2731 (2018). DOI:10.1126/sciadv.aat2731)

The need to share data effectively, and to compare simulations and test models, was illustrated clearly in the COVID-19 pandemic, which also demonstrated a willingness and commitment to data sharing across the international community (e.g. Amaro and Mulholland, J. Chem. Inf. Model. 2020, 60, 6, 2653-2656 https://doi.org/10.1021/acs.jcim.0c00319; Computing in Science & Engineering 2020, 22, 30-36 doi: 10.1109/MCSE.2020.3024155). There are important lessons to learn here, for simulations to be reproducible and reliable, for rapid testing, for exploiting data with machine learning, and for linking to data from other approaches. Tiemann et al. discuss how to develop these links, providing good perspectives and suggestions.

I agree completely with the statement of the authors that "Even if MD data represents only 1 % of the total volume of data stored in Zenodo, we believe it is our responsibility, as a community, to develop a better sharing and reuse of MD simulation files - and it will neither have to be particularly cumbersome nor expensive. To this end, we are proposing two solutions. First, improve practices for sharing and depositing MD data in data repositories. Second, improve the FAIRness of already available MD data notably by improving the quality of the current metadata."

This nicely states the challenge to the biomolecular simulation community. There is a clear need for standards for MD data and associated metadata. This will also help with the development of standards of best practice in simulations. The authors provide useful and detailed recommendations for MD metadata. These recommendations should contribute to discussions on the development of standards by researchers, funders, and publishers. Community organizations (such as CCP-BioSim and HECBioSim in the UK, BioExcel, CECAM, MolSSI, learned societies etc) have an important part to play in these developments, which are vital for the future of biomolecular simulation.

---

## [Author Response]

The following is the authors’ response to the original reviews.

**eLife assessment**
The study presents a valuable tool for searching molecular dynamics simulation data, making such data sets accessible for open science. The authors provide convincing evidence that it is possible to identify useful molecular dynamics simulation data sets and their analysis can produce valuable information.
**Public Reviews**

**Reviewer #1 (Public Review):**
Summary:Tiemann et al. have undertaken an original study on the availability of molecular dynamics (MD) simulation datasets across the Internet. There is a widespread belief that extensive, well-curated MD datasets would enable the development of novel classes of AI models for structural biology. However, currently, there is no standard for sharing MD datasets. As generating MD datasets is energy-intensive, it is also important to facilitate the reuse of MD datasets to minimize energy consumption. Developing a universally accepted standard for depositing and curating MD datasets is a huge undertaking. The study by Tiemann et al. will be very valuable in informing policy developments toward this goal.Strengths:The study presents an original approach to addressing a growing concern in the field. It is clear that adopting a more collaborative approach could significantly enhance the impact of MD simulations in modern molecular sciences.The timing of the work is appropriate, given the current interest in developing AI models for describing biomolecular dynamics.Weaknesses:The study primarily focuses on one major MD engine (GROMACS), although this limitation is not significant considering the proof-of-concept nature of the study.

We thank the reviewer for his/her comments. Moving forward, our plan includes expanding this research to encompass other MD engines used in biomolecular simulations and materials sciences, such as NAMD, Charmm, Amber, LAMMPS, etc. However, this requires parsing associated files to supplement the sparse metadata generally available for the related datasets

**Reviewer #2 (Public Review):**
Summary:Molecular dynamics (MD) data is deposited in public, non-specialist repositories. This work starts from the premise that these data are a valuable resource as they could be used by other researchers to extract additional insights from these simulations; it could also potentially be used as training data for ML/AI approaches. The problem is that mining these data is difficult because they are not easy to find and work with. The primary goal of the authors was to discover and index these difficult-to-find MD datasets, which they call the "dark matter of the MD universe" (in contrast to data sets held in specialist databases).The authors developed a search strategy that avoided the use of ill-defined metadata but instead relied on the knowledge of the restricted set of file formats used in MD simulations as a true marker for the data they were looking for. Detection of MD data marked a data set as relevant with a follow-up indexing strategy of all associated content. This "explore-and-expand" strategy allowed the authors for the first time to provide a realistic census of the MD data in non-specialist repositories.As a proof of principle, they analyzed a subset of the data (primarily related to simulations with the popular Gromacs MD package) to summarize the types of simulated systems (primarily biomolecular systems) and commonly used simulation settings.Based on their experience they propose best practices for metadata provision to make MD data FAIR (findable, accessible, interoperable, reusable).A prototype search engine that works on the indexed datasets is made publicly available. All data and code are made freely available as open source/open data.Strengths:The novel search strategy is based on relevant data to identify full datasets instead of relying on metadata and thus is likely to have many true positives and few false positives.The paper provides a first glimpse at the potential hidden treasures of MD simulations and force field parametrizations of molecules.Analysis of parameter settings of MD simulations from how researchers *actually* run simulations can provide valuable feedback to MD code developers for how to document/educate users. This approach is much better than analyzing what authors write in the Methods sections.The authors make a prototype search engine available.The guidelines for FAIR MD data are based on experience gained from trying to make sense of the data.Weaknesses:So far the work is a proof-of-concept that focuses on MD data produced by Gromacs (which was prevalent under all indexed and identified packages).As discussed in the manuscript, some types of biomolecules are likely underrepresented because different communities have different preferences for force fields/MD codes (for example: carbohydrates with AMBER/GLYCAM using AMBER MD instead of Gromacs).Materials sciences seem to be severely under-represented --- commonly used codes in this area such as LAMMPS are not even detected, and only very few examples could be identified. As it is, the paper primarily provides an insight into the *biomolecular* MD simulation world.The authors succeed in providing a first realistic view on what MD data is available in public repositories. In particular, their explore-expand approach has the potential to be customized for all kinds of specialist simulation data, whereby specific artifacts are used as fiducial markers instead of metadata. The more detailed analysis is limited to Gromacs simulations and primarily biomolecular simulations (even though MD is also widely used in other fields such as the materials sciences). This restricted view may simply be correlated with the user community of Gromacs and hopefully, follow-up studies from this work will shed more light on this shortcoming.The study quantified the number of trajectories currently held in structured databases as ~10k vs ~30k in generalist repositories. To go beyond the proof-of-principle analysis it would be interesting to analyze the data in specialist repositories in the same way as the one in the generalist ones, especially as there are now efforts underway to create a database for MD simulations (Grant 'Molecular dynamics simulation for biology and chemistry research' to establish MDDB' DOI 10.3030/101094651). One should note that structured databases do not invalidate the approach pioneered in this work; if anything they are orthogonal to each other and both will likely play an important role in growing the usefulness of MD simulations in the future.

We thank the reviewer for his/her comments. As mentioned to Reviewer 1, we intend to extend this work to other MD engines in the near future to go beyond Gromacs and even biomolecular simulations. Furthermore, as the value of accessing and indexing specialized MD databases such as MDDB, MemprotMD, GPCRmd, NMRLipids, ATLAS, and others has been mentioned by the reviewer, it is indeed one of our next steps to continue to expand the MDverse catalog of MD data. This indexing may also extend the visibility and widespreaded adoptability of these specific databases.

**Reviewer #3 (Public Review):**
Molecular dynamics (MD) simulations nowadays are an essential element of structural biology investigations, complementing experiments and aiding their interpretation by revealing transient processes or details (such as the effects of glycosylation on the SARS-CoV-2 spike protein, for example Casalino et al. ACS Cent. Sci. 2020; 6, 10, 1722-1734 https://doi.org/10.1021/acscentsci.0c01056) that cannot be observed directly. MD simulations can allow for the calculation of thermodynamic, kinetic, and other properties and the prediction of biological or chemical activity. MD simulations can now serve as "computational assays" Huggins et al. WIREs Comput Mol Sci. 2019; 9:e1393.https://doi.org/10.1002/wcms.1393. Conceptually, MD simulations have played a crucial role in developing the understanding that the dynamics and conformational behaviour of biological macromolecules are essential to their function, and are shaped by evolution. Atomistic simulations range up to the billion atom scale with exascale resources (e.g. simulations of SARS-CoV-2 in a respiratory aerosol. Dommer et al. The International Journal of High Performance Computing Applications. 2023; 37:28-44. doi:10.1177/10943420221128233), while coarse-grained models allow simulations on even larger length- and timescales. Simulations with combined quantum mechanics/molecular mechanics (QM/MM) methods can investigate biochemical reactivity, and overcome limitations of empirical forcefields (Cui et al. J. Phys. Chem. B 2021; 125, 689 https://doi.org/10.1021/acs.jpcb.0c09898).MD simulations generate large amounts of data (e.g. structures along the MD trajectory) and increasingly, e.g. because of funder mandates for open science, these data are deposited in publicly accessible repositories. There is real potential to learn from these data en masse, not only to understand biomolecular dynamics but also to explore methodological issues. Deposition of data is haphazard and lags far behind experimental structural biology, however, and it is also hard to answer the apparently simple question of "what is out there?". This is the question that Tiemann et al explore in this nice and important work, focusing on simulations run with the widely used GROMACS package. They develop a search strategy and identify almost 2,000 datasets from Zenodo, Figshare and Open Science Framework. This provides a very useful resource. For these datasets, they analyse features of the simulations (e.g. atomistic or coarse-grained), which provides a useful snapshot of current simulation approaches. The analysis is presented clearly and discussed insightfully. They also present a search engine to explore MD data, the MDverse data explorer, which promises to be a very useful tool.As the authors state: "Eventually, front-end solutions such as the MDverse data explorer tool can evolve being more user-friendly by interfacing the structures and dynamics with interactive 3D molecular viewers". This will make MD simulations accessible to non-specialists and researchers in other areas. I would envisage that this will also include approaches using interactive virtual reality for an immersive exploration of structure and dynamics, and virtual collaboration (e.g. O'Connor et al., Sci. Adv.4, eaat2731 (2018). DOI:10.1126/sciadv.aat2731)The need to share data effectively, and to compare simulations and test models, was illustrated clearly in the COVID-19 pandemic, which also demonstrated a willingness and commitment to data sharing across the international community (e.g. Amaro and Mulholland, J. Chem. Inf. Model. 2020, 60, 6, 2653-2656 https://doi.org/10.1021/acs.jcim.0c00319; Computing in Science & Engineering 2020, 22, 30-36 doi: 10.1109/MCSE.2020.3024155). There are important lessons to learn here, for simulations to be reproducible and reliable, for rapid testing, for exploiting data with machine learning, and for linking to data from other approaches. Tiemann et al. discuss how to develop these links, providing good perspectives and suggestions.I agree completely with the statement of the authors that "Even if MD data represents only 1 % of the total volume of data stored in Zenodo, we believe it is our responsibility, as a community, to develop a better sharing and reuse of MD simulation files - and it will neither have to be particularly cumbersome nor expensive. To this end, we are proposing two solutions. First, improve practices for sharing and depositing MD data in data repositories. Second, improve the FAIRness of already available MD data notably by improving the quality of the current metadata."This nicely states the challenge to the biomolecular simulation community. There is a clear need for standards for MD data and associated metadata. This will also help with the development of standards of best practice in simulations. The authors provide useful and detailed recommendations for MD metadata. These recommendations should contribute to discussions on the development of standards by researchers, funders, and publishers. Community organizations (such as CCP-BioSim and HECBioSim in the UK, BioExcel, CECAM, MolSSI, learned societies etc) have an important part to play in these developments, which are vital for the future of biomolecular simulation.

We thank the reviewer for his/her comments. Beyond the points mentioned to Reviewers 1 and 2, as the reviewer suggested, it would be of great interest to combine innovative and immersive approaches to visualize and possibly interact with the data collected. This is indeed more and more amenable thanks to technologies such as WebGL and programs such as Mol*, or even - as also pointed out by the reviewer - through virtual reality, for example with the mentioned Narupa framework or with the UnityMol software. For a comprehensive review on MD trajectory visualization and associated challenges, we refer to our recent review article https://doi.org/10.3389/fbinf.2024.1356659.

**Recommendations for the authors:**

**Reviewer #1 (Recommendations For The Authors):**
Some minor text editing would improve the readability of the manuscript.It would be very useful if the authors could share their perspectives on the best and most efficient approach to sharing datasets and code associated with a publication. My concern lies in the fact that Github, which is currently the dominant platform for sharing code, is not well-suited for hosting large MD datasets. As a result, researchers often need to adopt a workflow where code is shared on Github and datasets are stored elsewhere (e.g., Zenodo). While this is feasible, it adds extra work. Ideally, a transparent process could be developed to seamlessly share code and datasets linked to a study through a unified interface.

We thank the reviewer for this excellent suggestion. To our knowledge, there is yet no easy framework to jointly store and share code and data, linked to their scientific publication. Of course, code can be submitted to “generic” databases along with the data, but at the current state, those do not provide such useful features like collaborative work & track recording as done to the extent of GitHub.

Although GitHub is indeed a suitable platform to deposit code, we strongly advise researchers to archive their code in Software Heritage. In addition to preserving source code, Software Heritage provides a unique identifier called SWHID that unambiguously makes reference to a specific version of the source code.

So far, it is the responsibility of the scientific publication authors to link datasets and source codes (whether in GitHub or Software Heritage) in their paper, but also to make the reverse link from the data and code sharing platforms to the paper after publication.

As mentioned by the reviewer, a unified interface that could ease this process would significantly contribute to FAIR-ness in MD.

**Reviewer #2 (Recommendations For The Authors):**
L180: I am not aware that TRR files contain energy terms as stated here, my understanding was that EDR files primarily served that purpose.“…available in one dataset. Interestingly, we found 1,406 .trr files, Which contain trajectory but also additional information such as velocities, energy of the system, etc’ While the file is especially useful in terms of reusability, the large size (can go up to several 100GB) limits its deposition in most…”

Indeed, our formulation was ambiguous. The EDR files contain the detailed information on energies, whereas TRR files contain numerous values from the trajectory such as coordinates, velocities, forces and to some extent also energies (https://manual.gromacs.org/current/reference-manual/file-formats.html#trr).

L207: The text states that the total time was not available from XTC files, only the number of frames. However, XTC files record time stamps in addition to frame numbers. As long as these times are in the Gromacs standard of picoseconds, the simulation time ought to be available from XTCs.“…systems and the number of frames available in the files (Fig. 3-B). Of note, the frames do not directly translate to the simulation runtime - more information deposited in other files (e.g. .mdp files) is needed to determine the complete runtime of the simulation. The system was up…”.

Thank you for the useful comment, we removed this sentence. We now mention that studying the simulation time would be of interest in the future, especially when we will perform an exhaustive analysis of XTC files.

“Of note, as .xtc files also contain time stamps, it would be interesting to study the relationship between the time and the number of frames to get useful information about the sampling. Nevertheless, this analysis would be possible only for unbiased MD simulations. So, we would need to decipher if the .xtc file is coming from biased or unbiased simulations, which may not be trivial.”

Analysis of MDP files: Were these standard equilibrium MD or can you distinguish biased MD or free energy calculations?

Currently we do not distinguish between biased and unbiased MD, but in the future we may attempt to do so, e.g. by correlating it with standard equilibration force-fields/parameters, timesteps or similar. Nevertheless, a true distinction will remain challenging.

L336: typo: pikes -> spikes (or peaks?)“…simulations of Lennard-Jones models (Jeon et al., 2016). Interestingly, we noticed the appearance of several pikes at 400K, 600K and 800K, which were not present before the end of the year 2022. These peaks correspond to the same study related to the stability of hydrated crystals (Dybeck et al., 2023)’ Overall, thhis analysis revealed that a wide range of temperatures have been explored,…”

Thank you. We have corrected this typo.

Make clear how multiple versions of data sets are handled, e.g., if v1, v2, and v3 of a dataset are provided in Zenodo then which one is counted or are all counted?

We collected the latest version only of datasets, as exposed by default by the Zenodo API. To reflect this, we added the following sentence to the Methods and Materials section, Initial data collection sub-section:

“By default, the last version of the datasets was collected.”

L248 Analysis of GRO files seems fairly narrow because PDB files are very often used for exactly the same purpose, even in the context of Gromacs simulations, not the least because it is familiar to structural biologists that may be interested in representative MD snapshots. Despite all the shortcomings of abusing the PDB format for MD, it is an attempt at increased interoperability. Perhaps the authors can make sure that readers understand that choosing GRO for analysis may give a somewhat skewed picture, even within Gromacs simulations.

Thanks for this comment. We collected about 12,000 PDB files that could indeed be output from Gromacs simulations and easily be shared due to the universality of this format, but that could as well come from different sources (like other MD packages or the PDB database itself). We purposely decided to limit our study to files strictly associated with the Gromacs package, like MDP and XTC file types. However, we will extend our survey to all other structure-like formats and especially the PDB file type. We reflected this purpose in the following sentence (after line 281)

“Beyond .gro files, we would like to analyze the ensemble of the ~12,000 .pdb files extracted in this study (see Figure 2-B) to better characterize the types of molecular structures deposited.”

A simple template metadata file would be welcome (e.g., served from a GitHub/GitLab repository so that it can be improved with community input).

Thank you for this suggestion that we fundamentally agree with. However, the generation of such a file is a major task, and we believe that the creation of a metadata file template requires far-reaching considerations, therefore is beyond the scope of this paper and should not be decided by a small group of researchers. Indeed, this topic requires a large consensus of different stakeholders, from users, to MD program developers, and journal editors. It would be especially useful to organize dedicated workshops with representatives of all these communities to tackle this specific issue, as mentioned by Reviewer3 in his/her public review. As a basis for this discussion, we humbly proposed at the end of this manuscript a few non-constraining guidelines based on our experience retrieving the data.

To emphasize this statement, we added the following sentence at the end of the “Guidelines for better sharing of MD simulation data” section (line 420):

“Converging on a set of metadata and format requires a large consensus of different stakeholders from users, to MD program developers, and journal editors. It would be especially useful to organize specific workshops with representatives of all these communities to collectively tackle this specific issue.”

In "Data availability statement" it would be good to specify licenses in addition to stating "open source". Thank you for pointing out that GitLab/GitHub are not archives and that everyone should be strongly encouraged to submit data to stable archival repositories.

We added the corresponding licenses for code and data in the “Data availability statement” section.

**Reviewer #3 (Recommendations For The Authors)**
The paper is well written, with very few typographical or other minor errors.Minor points:Line 468-9 "can evolve being more user-friendly" should be "can evolve to being more user-friendly", I think.

Thank you, we have changed the wording accordingly.